# Regional Gray Matter Volume Changes in Parkinson’s Disease with Orthostatic Hypotension

**DOI:** 10.3390/brainsci11030294

**Published:** 2021-02-26

**Authors:** Jung Bin Kim, Hayom Kim, Chan-Nyung Lee, Kun-Woo Park, Byung-Jo Kim

**Affiliations:** 1Department of Neurology, Korea University Anam Hospital, Korea University College of Medicine, Seoul 02841, Korea; kjbin80@korea.ac.kr (J.B.K.); happyhy3@naver.com (H.K.); lcn001@naver.com (C.-N.L.); kunu@korea.ac.kr (K.-W.P.); 2BK21 FOUR Program in Learning Health Systems, Korea University, Seoul 02841, Korea

**Keywords:** MRI, voxel-based morphometry, Parkinson’s disease, orthostatic hypotension, visuoperception

## Abstract

Neurodegenerative change in the central nervous system has been suggested as one of the pathophysiological mechanisms of autonomic nervous system dysfunction in Parkinson’s disease (PD). We analyzed gray matter (GM) volume changes and clinical parameters in patients with PD to investigate any involvement in the brain structures responsible for autonomic control in patients with PD having orthostatic hypotension (OH). Voxel-based morphometry was applied to compare regional GM volumes between PD patients with and without OH. Multivariate logistic regression analysis using a hierarchical model was carried out to identify clinical factors independently contributing to the regional GM volume changes in PD patients with OH. The Sobel test was used to analyze mediation effects between the independent contributing factors to the GM volume changes. PD patients with OH had more severe autonomic dysfunction and reduction in volume in the right inferior temporal cortex than those without OH. The right inferior temporal volume was positively correlated with the Qualitative Scoring MMSE Pentagon Test (QSPT) score, reflecting visuospatial/visuoperceptual function, and negatively correlated with the Composite Autonomic Severity Score (CASS). The CASS and QSPT scores were found to be factors independently contributing to regional volume changes in the right inferior temporal cortex. The QSPT score was identified as a mediator in which regional GM volume predicts the CASS. Our findings suggest that a decrease in the visuospatial/visuoperceptual process may be involved in the presentation of autonomic nervous system dysfunction in PD patients.

## 1. Introduction

Orthostatic hypotension (OH) is a common and debilitating manifestation of autonomic dysfunction in patients with Parkinson’s disease (PD) [1,2]. While the compensation failure of sympathetic outflow for decreased venous return by standing up has been proposed as a mechanism of OH at the peripheral level [3], the pathophysiological mechanisms underlying OH in PD are still largely unknown at the central level. Pathological studies have demonstrated that neurodegenerative processes associated with α-synucleinopathies in PD occur along the central autonomic network, including the hypothalamus, dorsal motor nucleus of the vagus, and insular cortex [4,5,6]; this suggests that selective involvement of neurodegeneration in the brain structures responsible for autonomic control might be implicated in the pathophysiological mechanism underlying OH in PD patients.

In addition to the direct involvement of neurodegeneration in the central autonomic network, although the relationship is causative or associative is unclear, increasing evidence suggests that cognitive impairment could be associated with OH in patients with PD [7,8,9,10,11]. Given that OH and cognitive impairment often coexist as premotor signs in the early stage of PD [12,13], it is possible that OH and cognitive impairment may share pathophysiological mechanisms in the process of PD neurodegeneration. Understanding the mechanisms of the link between OH and cognitive function in PD may provide insight into the management strategy that could ameliorate subsequent cognitive decline in PD patients [13].

Few functional imaging studies found that several brain regions were associated with OH in patients with PD [14,15]. In a single-photon emission computed tomography study, perfusion in the bilateral anterior cingulate gyri was decreased in PD patients with OH relative to PD patients without OH [14]. An magnetic resonance imaging (MRI) study using arterial spin labeling found that more orthostatic blood pressure (BP) drops were correlated with lower perfusion to the right inferior frontal gyrus, right lingual gyrus, and precuneus in patients with alpha-synucleinopathies [15]. Although the aforementioned perfusion imaging studies provided insights into specific brain areas that may be potentially vulnerable to OH-induced recurrent cerebral hypoperfusion [14,15], little attention has been paid to exploring structural abnormalities that could be potential markers of OH in relation to cognitive decline in patients with PD.

Herein, we applied voxel-based morphometry to explore the presence of any regional gray matter (GM) changes associated with OH in patients with PD. We assumed there are regional GM changes in PD patients with OH; we aimed to investigate which clinical factors (e.g., autonomic dysfunction, cognitive impairment) are related to the regional GM changes, and to evaluate the independent effects of each clinical factor on the regional volume changes. We hypothesized that regional GM changes indicative of neurodegeneration could predict the severity of autonomic dysfunction including OH; this predictive model could be mediated by decreased cognitive function involved in compensating for orthostatic stress.

## 2. Materials and Methods

### 2.1. Subjects

We included 60 PD patients with Hoehn and Yahr stages from 1 to 4 who completed both MRI and autonomic function tests. All subjects underwent clinical MRI and MR angiography to confirm the absence of any structural lesions in the brain. PD was diagnosed using the diagnostic criteria from the United Kingdom PD Society Brain Bank [16]. Global neurocognitive function was evaluated using the Mini-Mental State Examination (MMSE) and Montreal Cognitive Assessment in patients with PD. Specific deficits in visuospatial/visuoperceptual and construction abilities were further assessed by the Qualitative Scoring MMSE Pentagon Test (QSPT, Appendix A) [17]. Motor function was assessed by using the Unified Parkinson’s Disease Rating Scale (UPDRS) Part III scores during the “off” period before autonomic function tests. Patients were categorized into having OH if they demonstrated a reduction in systolic or diastolic BP by at least 20 mmHg or 10 mmHg, respectively, within 3 min after standing up [18]. Written informed consent was obtained from all enrolled patients. This study was reviewed and approved by the institutional review board (No. 2019AN0418).

### 2.2. Autonomic Function Tests

All subjects were asked to discontinue any medications that could affect autonomic function, alcohol, and coffee for at least 24 h before the test. The autonomic function tests were performed during the “off” period in the following sequence: (1) quantitative sudomotor axon reflex test (QSART), (2) heart rate response to deep breathing, (3) Valsalva ratio, and (4) head-up tilt test (HUTT). The Composite Autonomic Severity Score (CASS), a measurement of the severity of autonomic dysfunction, was derived from the above-mentioned autonomic function tests [19]. Detailed methods for each test are described in our previous work [20,21].

### 2.3. MRI Acquisition and Voxel-Based Morphometry Analysis

MR images were acquired on a 3 T scanner (Prisma, Siemens Healthcare, Erlangen, Germany). Volumetric analysis was performed by acquiring a MPRAGE sequence using the following parameters: repetition time = 1790 ms, echo time = 2.51 ms, and voxel dimensions = 1 mm^3^. Data preprocessing and analysis were performed using SPM12 (http://www.fil.ion.ucl.ac.uk/spm, accessed on 25 February 2021), in which we impeded voxel-based morphometry with DARTEL [22]. Briefly, the preprocessing included (1) segmentation; (2) create a template; (3) normalizing to MNI space; and (4) smoothing the modulated GM volumes. Detailed procedures are described in our previous work [23].

Between-group comparisons of GM volume were assessed using an ANCOVA with age, sex, and total intracranial volume as nuisance variables. An absolute GM threshold of 0.2 was used to avoid possible edge effects around the border between GM and white matter. The thresholds for the statistical parametric maps were set at *p* < 0.001 and were uncorrected for multiple comparisons at a voxel level across the whole brain. The small-volume correction was further applied to the region found to be significant different between the groups using a 5 mm radius with a threshold of familywise error-corrected *p* < 0.05.

### 2.4. Statistical Analysis

Demographics and clinical variables were compared between PD patients with OH and those without OH using an independent t-test and chi-square test, where appropriate (*p* < 0.05). Univariate analysis was carried out to identify the effects of clinical variables on the voxel values extracted from the significant cluster in the between-group comparison using a simple linear regression. Multivariate analyses were conducted using a hierarchical linear regression to explore factors independently contributing to the severity of autonomic dysfunction: Model 1 controlling for age, sex, and UPDRS part III; Model 2 controlling for Model 1 variables + CASS; and Model 3 controlling for Model 2 variables + QSPT. To avoid the effect of multicollinearity between the variables, only variables for which the variance inflation factor was less than 5 entered into multivariate analyses as independent variables. Voxel values extracted from the significant cluster in the between-group comparison were correlated with the CASS and QSPT scores using partial correlation analysis, controlling for age (*p* < 0.05).

In order to examine the hypothetical path model, three regressions were conducted to determine the relationship between (1) path *a*: the independent variable (i.e., regional volume in the right inferior temporal cortex) and the mediator (i.e., QSPT); (2) path *b*: the mediator and the dependent variable (i.e., CASS); and (3) path *c* (direct effect): the independent variable and the dependent variable. According to the mediational procedure proposed by Baron and Kenny [24], the amount of indirect effect (*c’*) was calculated by the multiplication of the regression coefficients of path *a* and path *b*. If mediation is present, the magnitude and significance of coefficient *c’* should be less than c (direct effect). A Sobel test was conducted to determine the statistical significance of the proposed mediation pathway [25]. Statistical significance was set to *p* < 0.05. Statistical analyses were performed with the Statistical Package for the Social Sciences software (Version 21.0; IBM Corp., Armonk, NY, USA).

## 3. Results

Demographics and clinical data are presented in Table 1. PD patients with OH and those without OH showed no differences in demographics and PD-specific variables, including age, sex, proportion of hypertension, Hoehn and Yahr stage, UPDRS part III, MMSE, and Montreal Cognitive Assessment scores (all *p* > 0.05). PD patients with OH had higher CASS (5.1 ± 1.8 vs. 3.0 ± 1.7, *p* < 0.001) and lower QSPT scores (8.96 ± 3.1 vs. 10.8 ± 2.5, *p* = 0.012) than those without OH.

Voxel-based morphometry showed a significant reduction in regional GM volume in the posterior portion of the right inferior temporal cortex in PD patients with OH, compared with that in those without OH (MNI coordinate = 59/‒32/‒27, cluster size = 603 mm^3^, peak z score = 3.99, uncorrected voxel level *p* < 0.001, familywise error-corrected *p* < 0.001 after small-volume correction) (Figure 1A). No region demonstrated a significant increase in GM volume in PD patients with OH compared with PD patients without OH. Voxel values extracted from the right inferior temporal cortex cluster had a negative correlation with CASS (r = ‒0.434, *p* = 0.001) and a positive correlation with QSPT scores (r = 0.505, *p* < 0.001), after controlling for the possible effects of age (Figure 1B). Individual QSPT results and voxel values extracted from the right inferior temporal cortex cluster are presented in Figure 2.

Results of univariate analyses are summarized in Table 2. Decreasing UPDRS part III (*p* = 0.003) and CASS (*p* = 0.001), as well as increasing QSPT (*p* < 0.001) scores, were associated with an increase in regional GM volume in the right inferior temporal cortex. Results of multivariate analyses are summarized in Table 3. Decreasing CASS (*p* = 0.015) and increasing QSPT (*p* = 0.005) scores independently contributed to an increase in regional GM volume in the right inferior temporal cortex.

The results of the mediation analyses are shown in Figure 3. Regional volume in the right inferior temporal cortex negatively predicted CASS (direct effect; *β* = ‒0.431, *p* = 0.001). Then, regional volume in the right inferior temporal cortex positively predicted the QSPT scores (*β* = 0.509, *p* < 0.001). CASS was negatively predicted by the QSPT scores (*β* = ‒0.305, *p* = 0.018). Mediation analysis using Sobel test revealed the significance of mediation pathway by showing a reduction of direct effect of regional volume in the right inferior temporal cortex on CASS (*z* = ‒2.145, *p* = 0.003).

## 4. Discussion

We observed a reduction in the regional GM volume in the posterior portion of the right inferior temporal cortex in PD patients with OH compared to those without OH. In addition, we observed that a smaller volume in the right inferior temporal cortex was correlated to more severe autonomic dysfunction (higher CASS) and visuospatial/visuoperceptual impairment (lower QSPT) in patients with PD. It is notable that QSPT score was found to be a more potent independent contributing factor than CASS for regional GM volume change in the right inferior temporal cortex in PD patients. Moreover, CASS could be predicted by the regional GM volume, and the presence of mediation effect of QSPT score was found in the prediction model.

The inferior temporal cortex has been known to affect visuoperceptual processing by linking the visual information from the visual cortex to both the limbic system and the frontal lobe [26]. Furthermore, it is imperative to activate the inferior temporal cortex to reinstate visual memory immediately before making a decision [27]. Considering the functional role of the inferior temporal cortex in terms of visuoperception, our results imply that impaired visuoperceptual responses to sudden view changes while standing up might be related to the pathophysiological mechanism of OH in PD patients. Meanwhile, several lines of evidence have shown that PD patients with OH are susceptible to cognitive impairments in visuospatial memory and visuoperceptual process compared to those without OH [28]. Given the domain-specific cognitive impairments in visuoperception in PD with OH [28], we speculate that our finding of the reduction in regional GM volume in the right inferior temporal cortex might be associated with visually-mediated mechanisms underlying OH in patients with PD. We applied the QSPT method to measure visuospatial/visuoperceptual function by subdividing the total score of intersecting pentagons copying test from 0 to 13 [17]. The correlation between right inferior temporal volume reduction and decreased QSPT scores, as well as independent contributing effect of QSPT scores on the regional volume, could support our speculation that the volume change in the right inferior temporal cortex was clinically relevant to changes in visually-mediated cognitive function.

An experimental study demonstrated that visually induced tilt illusions could elicit autonomic cardiovascular responses resembling their initial responses to passive head-up tilting [29]. These results suggest that not only baroreflex but also visual input per se may contribute to the initial autonomic cardiovascular responses to postural changes. Based on the notion regarding visually-induced initiation of autonomic reflex, we hypothesize that GM volume reduction in the right inferior temporal cortex may be associated with a decreased function of the visuoperceptual process involved in autonomic cardiovascular responses to compensate for the drop in orthostatic blood pressure. Our finding of the negative relationship between the GM volume of the right inferior temporal cortex and the severity of autonomic dysfunction (i.e., CASS) could support our speculation that the right inferior temporal cortex might be an anatomical correlate of impaired autonomic function in patients with PD. Moreover, our findings of mediation analysis further support our hypothesis that decreased function of the visuoperceptual process might be implicated in the mechanism underlying impairment of autonomic compensation in response to orthostatic stress, leading to OH.

The presence of hemispheric lateralization of cardiovascular autonomic control is well-known. Intracarotid amobarbital injection in patients with epilepsy showed sympathetic activation after left hemisphere injection and parasympathetic activation after injection to right hemisphere [30]. Moreover, we observed that patients with neurocardiogenic syncope had right insular atrophy, and smaller right insular volumes were related to the magnitude of BP drop during the HUTT [31]. Taken together, volume reduction in the right temporal lobe in PD patients with OH, compared to that in those without OH, in the present study could be interpreted as an impaired sympathetic outflow to compensate for the drop in blood pressure by standing up.

There are some limitations in our study. First, sample size in our population was relatively small. Second, the current study is cross-sectional; therefore, the interpretation of our results is limited with respect to a causal relationship. Further prospective studies incorporating a longitudinal design may provide insight into the causal relationship between right inferior temporal atrophy and disease progression. In addition, further prospective studies with larger populations are needed to generalize our results. Finally, our voxel-based morphometry result may be limited by the use of a lenient exploratory threshold.

## 5. Conclusions

We observed a GM volume reduction in the right inferior temporal cortex in PD patients with OH, implicating a role of the right inferior temporal cortex in the pathophysiological mechanism underlying OH among PD patients. Together with previous knowledge, we suggest that GM volume reduction in the right inferior temporal cortex may be associated with a decrease in the visuoperceptual process involved in eliciting the initial autonomic responses to postural changes, resulting in OH in PD patients.

## Figures and Tables

**Figure 1 brainsci-11-00294-f001:**
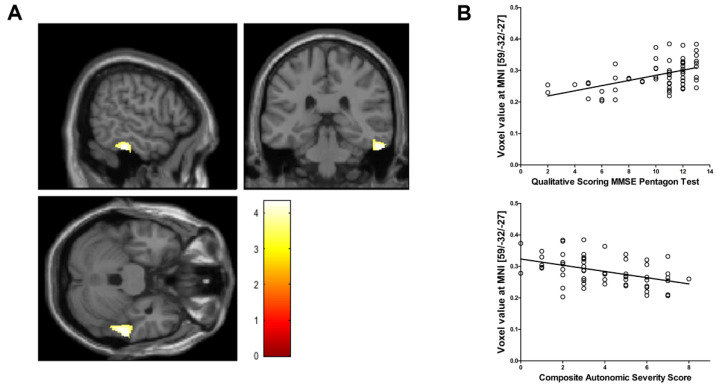
(**A**) Voxel-based morphometry shows a significant regional gray matter volume reduction in the right inferior temporal cortex in Parkinson’s disease patients with orthostatic hypotension (OH) compared with those without OH (voxel-level uncorrected *p* < 0.001, familywise error-corrected *p* < 0.001 after small volume correction). The color bar represents the T value. (**B**) The normalized volume of the right inferior temporal cortex (raw volume/estimated total intracranial volume [eTIV] × 100) has a positive relationship with Qualitative Scoring MMSE Pentagon Test (QSPT) and a negative relationship with Composite Autonomic Severity Score (CASS).

**Figure 2 brainsci-11-00294-f002:**
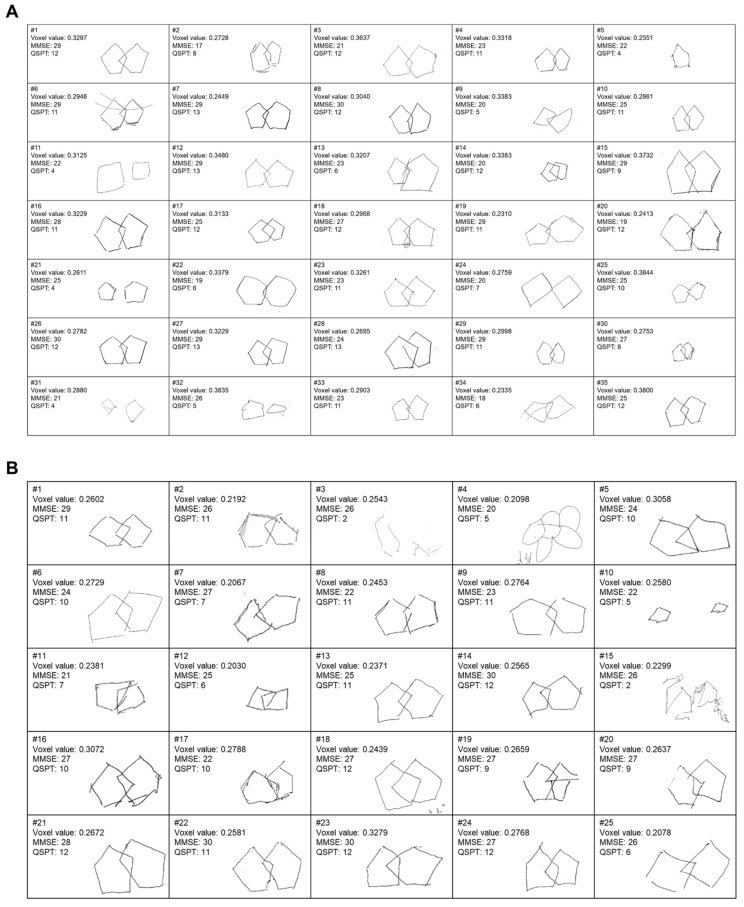
Individual results of Mini-Mental State Exam (MMSE), Qualitative Scoring MMSE Pentagon Test (QSPT), and voxel values extracted from the right inferior temporal cortex cluster in Parkinson’s disease patients without orthostatic hypotension (OH) (**A**) and those with OH (**B**).

**Figure 3 brainsci-11-00294-f003:**
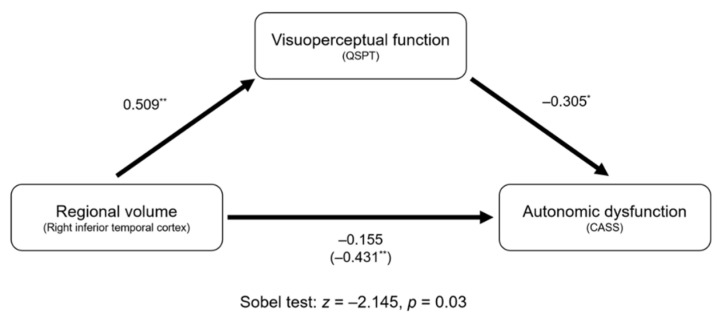
Mediation analysis for autonomic dysfunction. Unstandardized coefficients are presented. The number in parenthesis (i.e., ‒0.431) indicates the direct effect of regional volume in the right inferior temporal cortex on the autonomic dysfunction. Lesser indirect effect coefficient (i.e., ‒0.155) than direct coefficient indicates the presence of mediation effect. * *p* < 0.05, ** *p* < 0.01.

**Table 1 brainsci-11-00294-t001:** Demographic and clinical characteristics of the patients.

	PD with OH (*n* = 25)	PD without OH (*n* = 35)	*p* Value
Age, years	75.8 ± 7.7	74.6 ± 4.0	0.489
Male, *n* (%) ^a^	15 (60.0)	17 (48.6)	0.382
Hypertension, *n* (%) ^a^	13 (52.0)	18 (51.4)	0.965
H&Y stage	2.3 ± 0.6	2.3 ± 0.6	0.676
UPDRS part III	27.8 ± 10.9	23.6 ± 9.3	0.108
CASS	5.1 ± 1.8	3.0 ± 1.7	<0.001
MMSE	25.6 ± 2.9	24.6 ± 3.9	0.238
QSPT	8.96 ± 3.1	10.8 ± 2.5	0.012
MoCA	19.9 ± 4.9	19.6 ± 6.1	0.805

PD: Parkinson’s disease, OH: orthostatic hypotension, H&Y: Hoehn & Yahr, UPDRS: Unified Parkinson’s Disease Rating Scale, CASS: composite autonomic severity score, MMSE: Mini-Mental State Exam, QSPT: Qualitative Scoring MMSE Pentagon Test, MoCA: Montreal Cognitive Assessment. Each value represents the mean ± standard deviation. Independent *t*-test was used to compare the variables. ^a^ Chi-square test was performed.

**Table 2 brainsci-11-00294-t002:** Results of univariate analyses.

Variables	*β*	*p* Value
Age	‒0.001	0.164
Male sex	‒0.002	0.872
H&Y	‒0.008	0.421
UPDRS part III	‒0.002	0.003
CASS	‒0.010	0.001
MMSE	0.001	0.768
QSPT	0.008	<0.001
MoCA	0.001	0.294

H&Y: Hoehn & Yahr, UPDRS: Unified Parkinson’s Disease Rating Scale, CASS: composite autonomic severity score, MMSE: Mini-Mental State Exam, QSPT: Qualitative Scoring MMSE Pentagon Test, MoCA: Montreal Cognitive Assessment.

**Table 3 brainsci-11-00294-t003:** Results of multivariate analyses.

Variables	Model 1	Model 2	Model 3
Age	NS	NS	NS
Male sex	NS	NS	NS
UPDRS part III	‒0.357 **	‒0.285 *	NS
CASS		‒0.374 **	‒0.284 *
QSPT			0.367 **

Values represent the regression coefficient (*β*). Hierarchical linear regression analysis: Model 1 adjusted for age, sex, UPDRS part III; Model 2 adjusted for Model 1 variables + CASS; Model 3 adjusted for Model 2 + QSPT. * *p* < 0.05, ** *p* < 0.01. UPDRS: Unified Parkinson’s Disease Rating Scale, CASS: composite autonomic severity score, QSPT: Qualitative Scoring MMSE Pentagon Test.

## Data Availability

Data available at request.

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
