# Peer review of "Regional Gray Matter Volume Changes in Parkinson’s Disease with Orthostatic Hypotension"

_brainsci, 2021, doi:10.3390/brainsci11030294_

Round 1
Reviewer 1 Report
This manuscript reports a study which has investigated autonomic dysfunction in patients with Parkinson’s disease and its correlated with regional gray matter volumetry. Authors found an association between autonomic dysfunction and gray matter volume of the right inferior temporal lobe. They also found an association between right inferior temporal volume and visuospatial dysfunction assessed by score the pentagons of the MMSE test.
The findings of the study are original and new and propose new insight to the understanding of autonomic dysfunction in PD. The methods are appropriate and relevant, particularly the investigation of autonomic function. Results and Discussion are clearly expressed, including the limitation part.
Minor point
In the Figure 3 why do the authors present two coefficients (-0.155 and -0.431) for the association between the volume of the right inferior temporal lobe and the score of autonomic dysfunction?
Author Response
- Minor point
In the Figure 3 why do the authors present two coefficients (-0.155 and -0.431) for the association between the volume of the right inferior temporal lobe and the score of autonomic dysfunction?
Reply) Thank you for pointing out what we have overlooked. There was an insufficient description in the figure caption. The number in parenthesis (i.e., -0.431) represents the coefficient of path c that is a direct effect of the independent variable to the dependent variable. Another number, '-0.155' is the coefficient of path c' that is an indirect effect. Since the presence of mediation effect, the magnitude and significance of coefficient c' were less than c. We clarified the figure caption as following:
"The number in parenthesis (i.e., ‒0.431) indicates the direct effect of regional volume in the right inferior temporal cortex on the autonomic dysfunction. Lesser indirect effect coefficient (i.e., ‒0.155) than direct coefficient indicates the presence of mediation effect." (page 6, line 343-345)
Reviewer 2 Report
Kim et al perfomed a MRI study with voxel-based morphometry analysis, observing grey matter volume reduction in the right inferior temporal cortex of PD patients presenting orthostatic hypotension (OH).
Its impaired function, associated with reduced visuo-perceptual processes, has been suggested to be involved in the initial autonomic response to postural changes.
This is a well designed study, that despite the small cohort provides interesting information.
The diagnosis of OH, made using head up tilt test instead of passive standing, is particularly important, given the impact on patients’ daily life.
In light of this, it would be interesting to study more specifically some other aspects such as the magnitude of the BP drop and the presence of symptomatic OH patients vs asymptomatic OH patients, and their possible correlations with imaging data.
More solid data should be obtained from a prospective study on a larger cohort.
Misspelling (deceasing CASS) page 7, line 240.
Author Response
- In light of this, it would be interesting to study more specifically some other aspects such as the magnitude of the BP drop and the presence of symptomatic OH patients vs asymptomatic OH patients, and their possible correlations with imaging data.
Reply) Thank you for your valuable recommendation. We did not find any relationship between the difference of BP during head-up tilt and regional volume in the right inferior temporal cortex (systolic BP: r = 0.048, p = 0.819; diastolic BP: r = 0.141, p = 0.502). Since patients were quickly returned to the supine position from head-up tilt after symptom occurrence or excessive BP drop, it is plausible that the BP difference during the head-up tilt test may not fully reflect the magnitude of the BP drop. We also did not find any difference in regional volume in the right inferior temporal cortex between patients with symptomatic OH (n = 11) and those with asymptomatic OH (n = 14) (0.2417 ± 0.0350 vs. 0.2651 ± 0.0266. p = 0.069). Although symptomatic OH patients tended to have a smaller volume in the right inferior temporal cortex than asymptomatic OH patients, they were not statistically significant. It might be attributed to the small sample size that was not enough to perform within-group subanalysis. Further studies with larger populations may reveal the possible difference in the regional volume between the symptomatic and asymptomatic OH patients.
- More solid data should be obtained from a prospective study on a larger cohort.
Reply) We agree with your opinion that further prospective studies on a larger cohort should be done to generalize our results. We added the necessity of further prospective studies with larger populations in the limitation section of the revision.
Please see the following sentences:
"In addition, further prospective studies with larger populations are needed to generalize our results." (page 8, line 422-423)
- Misspelling (deceasing CASS) page 5, line 320.
Reply) We corrected the misspelling.